# HDR-EfficientNet: A Classification of Hypertensive and Diabetic Retinopathy Using Optimize EfficientNet Architecture

**DOI:** 10.3390/diagnostics13203236

**Published:** 2023-10-17

**Authors:** Qaisar Abbas, Yassine Daadaa, Umer Rashid, Muhammad Zaheer Sajid, Mostafa E. A. Ibrahim

**Affiliations:** 1College of Computer and Information Sciences, Imam Mohammad Ibn Saud Islamic University (IMSIU), Riyadh 11432, Saudi Arabia; ymdaadaa@imamu.edu.sa (Y.D.);; 2Department of Computer Science, Quaid-i-Azam University, Islamabad 44000, Pakistan; umerrashid@qau.edu.pk; 3Department of Computer Software Engineering, MCS, National University of Science and Technology, Islamabad 44000, Pakistan; 4Department of Electrical Engineering, Benha Faculty of Engineering, Benha University, Benha 13518, Qalubia, Egypt

**Keywords:** diabetic retinopathy, hypertensive retinopathy, deep learning, transfer learning, convolutional neural network, inception model

## Abstract

Hypertensive retinopathy (HR) and diabetic retinopathy (DR) are retinal diseases closely associated with high blood pressure. The severity and duration of hypertension directly impact the prevalence of HR. The early identification and assessment of HR are crucial to preventing blindness. Currently, limited computer-aided methods are available for detecting HR and DR. These existing systems rely on traditional machine learning approaches, which require complex image processing techniques and are often limited in their application. To address this challenge, this work introduces a deep learning (DL) method called HDR-EfficientNet, which aims to provide an efficient and accurate approach to identifying various eye-related disorders, including diabetes and hypertensive retinopathy. The proposed method utilizes an EfficientNet-V2 network for end-to-end training focused on disease classification. Additionally, a spatial-channel attention method is incorporated into the approach to enhance its ability to identify specific areas of damage and differentiate between different illnesses. The HDR-EfficientNet model is developed using transfer learning, which helps overcome the challenge of imbalanced sample classes and improves the network’s generalization. Dense layers are added to the model structure to enhance the feature selection capacity. The performance of the implemented system is evaluated using a large dataset of over 36,000 augmented retinal fundus images. The results demonstrate promising accuracy, with an average area under the curve (AUC) of 0.98, a specificity (SP) of 96%, an accuracy (ACC) of 98%, and a sensitivity (SE) of 95%. These findings indicate the effectiveness of the suggested HDR-EfficientNet classifier in diagnosing HR and DR. In summary, the HDR-EfficientNet method presents a DL-based approach that offers improved accuracy and efficiency for the detection and classification of HR and DR, providing valuable support in diagnosing and managing these eye-related conditions.

## 1. Introduction

In the United States, hypertension affects approximately 9.5 million individuals [1], which is anticipated to grow. It is a common, universal ailment. The retina and the retinal arteries undergo many alterations due to the rise in blood pressure, or HR. Early HR identification is crucial as it can increase cardiovascular risk and retinal microcirculation. These two HR-related diseases have been identified in a large population of hypertensive people. As HR symptoms [2] appear, many people experience visual loss. Recent research has shown that retinal microvascular changes may be seen using a fundus digital camera. This fundus camera is utilized to non-invasively screen many HR patients since it is affordable [3] and easy to use, and most anatomical characteristics, as shown in Figure 1, of lesions are evident in this form of imaging.

The most common form of eye disease, which has recently spread around the globe, is hypertension caused by an increase in artery pressure [4], which can harm several human organs, including the kidneys, heart, and retina [5]. Of all these effects, cardiovascular disease, which results in mortality, is mainly brought on by HR [6]. In general, HR is a condition that affects the retina and is brought on by an excessive increase in blood pressure. The appearance of symptoms, including bleeding spots in the retina, cotton wool patches, and micro-aneurysms, is a definite sign of eye disease. Early detection and proper treatment of ocular disorders associated with HR can save lives [7].

An ophthalmologist, a medical expert, uses a non-invasive, cost-effective analysis of microscopic retinal pictures to determine whether HR disease is present. The primary goal of automated systems [8] is to quickly and easily assess the existence of HR while removing the burden of lengthy image assessments from ophthalmologists. Many studies have already developed methods for segmenting retinal arteries and HR lesions, extracting characteristics, and, most recently, developing supervised machine learning classifiers for HR [9] eye-related disease.

The increased presence of retinal veins lowers the artery-to-vein diameter (i.e., the A/V ratio, where A is the total diameter of all arteries and V is the total diameter of all veins in the image) and is an important HR indicator. Using an image analysis method to identify HR-related eye illnesses makes it challenging to detect the diameter of retinal vessels [10] and other parameters like AVR. Additionally, it might be difficult for ophthalmologists to obtain accurate measurements of vascular diameters. Ophthalmologists diagnose HR by looking for anomalies in retinographics and automatically analyzing digital fundus photographs. As mentioned, these anomalies are indicative of damage to essential parts of the eye and may escalate to HR if they are not caught in time. In addition, whereas DR-related eye illness cannot be reversed, HR-related eye disease may be [11]. Some retinal disorders, including eye conditions connected to HR, can be diagnosed by ophthalmologists using computer-aided diagnostic (CAD) systems. These tools facilitate immediate diagnosis, which benefits research and the healthcare sector. Ophthalmologists employ these technologies to identify and treat eye disorders, especially those that are HR-related. In this paper, we have modified the EfficientNet architecture to develop this CAD system.

EfficientNet stands out as a compelling choice among DL models due to its unique scalability, efficiency, and performance combination. Unlike other models that often require manual adjustments to balance complexity and resource constraints, EfficientNet’s compound scaling approach uniformly scales the network across depth, width, and resolution dimensions. This leads to optimal model size selection, enabling the architecture to be tailored precisely for the task. Its efficient use of parameters translates to faster training times and reduced computational demands, making it particularly suitable for resource-constrained environments such as mobile devices. Furthermore, EfficientNet incorporates attention mechanisms that enhance its ability to capture subtle features within medical images, such as retinography scans, which is crucial for accurate disease detection. In this paper, the HDR-EfficientNet model demonstrates the adaptability and efficacy of EfficientNet by addressing the challenges of early-stage eye disease diagnosis while efficiently utilizing available computational resources.

### 1.1. Clinical Importance

The clinical importance of this CAD system lies in its ability to provide accurate, efficient, and objective classification of hypertensive retinopathy (HR) and diabetic retinopathy (DR). It supports clinicians in making informed decisions, enhances patient care, and contributes to improved disease management and prevention as described in Table 1. Table 1 highlights the clinical significance of the HDR-EfficientNet system in classifying HR and DR eye-related diseases. The system promotes early disease detection, ensuring swift interventions and halting disease progression. It offers consistent, objective diagnostics by minimizing subjective interpretation. Its rapid retinograph image analysis speeds up diagnosis. It is instrumental in optimal healthcare resource allocation, remote assessments for telemedicine, and lightens clinician workload. Additionally, it supports population screenings, widens access to specialist care, bolsters research, and educates patients, enhancing overall healthcare efficacy and patient understanding.

### 1.2. Paper Organization

The remainder of this paper is structured as follows: The background and related studies are discussed in Section 2 and Section 3, respectively. Data acquisition and results are described in Section 4. Discussions are presented in Section 5 with future works. Finally, the paper concludes in Section 6.

## 2. Background

Feature extraction and object identification are areas where previous systems have heavily used deep learning models (DLMs) [8,9,10,11]. A unique active deep learning-based convolutional neural network (CNN) was developed to deal with the challenges of multilayered network design training. In practice, training the CNN architecture is straightforward. The CNN model has also been extensively utilized in studies aimed at pervasive image recognition. The CNN model outperforms other models for use with limited training datasets compared to their performance. In an earlier study, the scientists claimed that the CNN model automatically picked up unique characteristics from the unprocessed samples.

It is important to note that while these patterns are characteristic of DR and HR, as shown in Table 2, there can be variations in presentation based on the severity of the disease and individual patient factors. Additionally, retinograph images provide a valuable tool for identifying these patterns and assisting in accurately diagnosing and managing both conditions. Diabetic retinopathy (DR) and hypertensive retinopathy (HR) are two distinct retinal diseases that can be diagnosed through retinograph images. While both conditions affect the blood vessels in the retina, there are specific patterns that differentiate them. Here are some critical differences in the patterns observed in retinograph images for diagnosing DR and HR.

Although several machine learning (ML) techniques have been used to categorize retinal fundus pictures into normal and HR, these algorithms still need significant help. It is challenging to find and identify essential elements for HR symptoms from retinography photographs to define HR quality without using laborious pre- or post-image processing techniques. It is difficult to train and evaluate the network since there are not enough datasets classified as “normal” and “high risk” by a qualified medical professional. Automated methods thus have a better habit of identifying sickness. The accuracy of the researchers’ models was assessed against the body of literature while their networks were being trained using hand-crafted features. As a result, rather than depending on hand-crafted characteristics, determining the top traits requires an automated process. While several models that will surely learn features via deep learning are created, they all share the same weights at every layer. Then, it becomes more challenging for layers to communicate weight parameters to the network’s later stages so that they can make informed choices.

EfficientNet follows a hierarchical structure [12], where the model architecture is scaled uniformly across three dimensions: depth, width, and resolution. The scaling is controlled by a coefficient called “compound scaling,” which determines the size of the network. At a high level, the EfficientNet architecture consists of a series of blocks repeated multiple times. These blocks typically comprise convolutional layers, activation functions (such as Swish), and normalization layers (such as batch normalization). The specific architecture of each EfficientNet variant, from B0 to B7, involves stacking multiple blocks and increasing the number of channels and layers as you move toward more significant variants. The number of channels controls the width of the network, while the number of layers determines its depth.

Additionally, EfficientNet incorporates other techniques, such as squeeze-and-excitation (SE) blocks, which selectively emphasize informative features, and a global average pooling layer to aggregate information from different spatial locations. It is important to note that the exact architecture and configuration can vary based on the implementation and specific requirements of the task at hand. Suppose you are looking for a visual representation of the EfficientNet architecture. In that case, refer to the original research papers on EfficientNet, which often include diagrams illustrating the model’s structure and building blocks.

Using a DL model for fully automated retinography image processing, particularly for HR illnesses, is the primary objective of this project. This attempt extracts clinical information without image processing or conventional machine learning methods, which call on domain-expert knowledge. A HR diagnostic system will be implemented using multilayer HDR-EfficientNet. The training of the HDR-EfficientNet model used several HR photos with anatomical details to successfully detect HR. This work suggests a transfer-learning mechanism for using fundus pictures to recognize unique anomalies. Although specialists have put a lot of effort into categorizing eye-related diseases, diagnosing sickness in its early stages is still challenging since infected and healthy eye regions have many comparable features. Recognition is made more difficult by various plant leaf types, lighting and brightness changes, distortion and blurring in the processed photographs, and many other elements. As a result, there is still room for advancement in the accuracy and computer power of diagnosing HR and DR eye-related diseases. In the study that was just given, we tried to address the issue of classifying eye-related diseases by putting out a proper DL strategy called EfficientNet. We modified the existing EfficientNet-v2 model by adding an attention mechanism (AM) and additional thick layers at the end of the framework structure. The suggested HDR-EfficientNet technique successfully extracts high-level indicators of diseased areas and correlates them with associated groups using an end-to-end training process. The AM method also increases the memory power of the suggested remedy by conveying pertinent information about features that cannot be ticked, such as unhealthy patches of DR and HR images.

## 3. Literature Review

According to this literature review, a segmentation-based algorithm should be used to identify hypertensive retinopathy from fundus pictures [13]. After identifying different HR-related traits, the researchers used an ML classification approach to identify HR from retinographics in their trials. The papilledema indicators, the index of tortuosity, the location of the optic disc (OD), the mean fractal dimension (mean-D), and the artery-to-vein diameter ratio (A/VR) are manually crafted features that are used in automated approaches to identify retinal irregularities such as graded HR and vascular bifurcation [14,15,16,17,18].

The scientists used a cutting-edge technique in their study [19] to recognize HR-related eye illnesses. The researchers discovered cotton wool patches (CWS), one of the most significant diagnostic clues for HR illness. Researchers enhanced candidate areas with the Gabor filter bank before binarizing the image with an adaptive threshold technique. To categorize the numerous retinal blood vessel types, ref. [20] used a multi-layered neural network containing invariant moment markers, Gabor coefficients, and wavelet coefficients. They achieved noticeably improved segmentation outcomes on the DRIVE dataset.

In [21], the authors outline a nine-step automated process for extracting the OD region, segmenting vessels, detecting color features, calculating the A/V ratio, identifying veins and arteries, calculating the AM/FM characteristics as well as the mean RED intensity, and finally classifying the images as HR or standard fundus photos. For a set of 74 color fundus images, a 0.84 AUC was reached, along with 90% sensitivity and 67% specificity. The optic disc, blood vessels, hemorrhages, macula, exudates, and drusen were among the retinal changes identified in the retinography photographs by an ICA (independent component analysis) on a wavelet sub-band [22]. On 50 photos, this method was tested, and the outcomes were accurate.

As opposed to [23], where the authors proposed a technique for identifying HR using traits from previously examined color retinography pictures, the retinal vessels could be seen more clearly thanks to applying CLAHE to convert the retinography pictures into green-channel images. Second, it eliminated the optic disk via morphological closure. The backdrop was destroyed using subtraction. Then, utilizing zoning, attributes were extracted. A neural network with backpropagation was ultimately employed as a classifier. The degree of accuracy was 95%. The authors of [24] also provide a method for segmenting retinal arteries from fundus pictures using the ELM classifier—the classifier’s input feature vector comprised 39 local, morphological, and other attributes. On the DRIVE dataset, this approach has 96% accuracy, 71.4% sensitivity, and 98.6% specificity. Any learning-based procedure may categorize fundus photos, but each method takes a different approach. These techniques reduce the amount of pre-processing conducted to the fundus. Deep-learning architectures can directly complete many image-processing tasks, such as segmentation and feature extraction.

In [25], the researchers utilized an architecture combining the random Boltzmann machine (RBM) and deep neural network (DNN) techniques to detect changes in arterial blood vessels. They introduced unique characteristics in the deep learning method by utilizing the arteriovenous ratio (AVR) and the optic disc (OD) region. The research demonstrated significantly improved accuracy compared to previous methods. The authors of [26] presented a CNN-based technique for extracting the optic disc, retinal arteries, and fovea centralis. The CNN architecture consisted of seven layers, facilitating the identification and segmentation of these critical retinal structures. These studies highlight the application of deep learning methods, specifically CNNs and the integration of RBM and DNN, for recognizing and detecting HR-related abnormalities in retinal images. Each approach contributes unique features and techniques to enhance the accuracy and effectiveness of identifying hypertensive retinopathy. The fovea centralis, OD, the retinal arteries, and the retinal backdrop were all represented by four nodes in the output of their CNN design. A mean classification accuracy of 92.6% on the DRIVE dataset was attained. The retinal vasculature was divided and classified into arteries and veins by several researchers using CNN [27,28,29]. These methods have good accuracy: 88.9% for a dataset of 100 low-quality photos and 93%+ for the DRIVE dataset. Lastly, the authors in [30] proposed an automatic CNN method for finding exudates in retinography images. All of the characteristics were gathered throughout CNN’s training process. CNN was provided with patches of varying sizes, with the processed pixel at the patch’s center. Convolutional layers were utilized to determine whether each pixel was a component of an exudate. There are no exudates in the OD area. Therefore, it is not included. In addition to the input and output layers, the CNN design includes four convolutional and pooling layers. The method has a 77% F-score and identical performance metrics for the DRiDB dataset: sensitivity and positive predictive value (PPV). Table 3 describes the limitations of the state-of-the-art approaches.

## 4. Materials and Methods

The detection and classification of HR and DR eye-related diseases are the focus of our proposed study, which is based on the EfficientNet method known as enhanced EfficientNetV2. The HR and DR datasets, which contain 12,000 photos of retinograph images overall, were used to test and evaluate the performance of the suggested approach. We used data augmentation techniques to balance this dataset within each class. By adding more layers to the model’s base, this suggested study aims to improve the EfficientNet technique for identifying and categorizing retinal fundus diseases. The model’s performance was intended to be improved by enabling it to recognize more intricate patterns and characteristics in the photos. An extensive collection of pictures of retinal fundus images, including normal, HR, and DR, was used to train the improved model, which was given the name enhanced EfficientNetV2.

To help ophthalmologists and clinical experts increase the accuracy of detection and decrease losses, the proposed work has the potential to help build more precise and trustworthy models for identifying and categorizing eye-related diseases. The performance of the upgraded model is increased by the inclusion of layers at its base. Figure 2 depicts the whole flow of our modified model.

### 4.1. Acquisition and Preparation of Dataset

To enhance the HDR-EfficientNet Diagnostic System, we aimed to create a dataset containing enough retinography images. As a result, we gathered 6000 normal retinography images and HR of 700 and DR of 700 images to compare different diagnostic approaches using HDR-EfficientNet. In total, we utilized 7400 retinographs. The distribution of these datasets is described in Table 4.

There are several publicly available datasets for experiments related to hypertensive and diabetic retinopathy. These datasets contain retinal images that have been annotated to identify the presence and severity of these conditions. Keep in mind that new datasets might have emerged since then, so it is a good idea to search for the most up-to-date resources.

The manual differentiation of HR and regular fundus images from distinct datasets required the expertise of a skilled ophthalmologist to create the training dataset. To establish a gold standard, the ophthalmologist evaluated all HR-related parameters in a collection of 7400 fundus photographs, as depicted in Figure 3. This dataset formed the basis for training and testing the HDR-EfficientNet system. The dataset creation process involved the utilization of three distinct datasets, each serving a specific purpose. The training and testing datasets were carefully formed by manually distinguishing HR and regular fundus images from these datasets. To ensure consistency, all images were resized to 300 × 300 pixels for analysis. These fundus images were obtained as part of a standard procedure for identifying individuals with hypertension. The compilation of HR, normal, and DR datasets for ground truth evaluation required the expertise of a competent ophthalmologist. A visual representation of the fundus database used in the investigation is illustrated in Figure 3. To increase the size of the training and testing datasets, a preprocessing phase was employed, which involved data augmentation techniques. These techniques helped to augment the dataset by generating additional variations of the available images, enhancing the model’s ability to generalize and improve performance. By carefully curating and augmenting the dataset, we aimed to enhance the training and testing process of the HDR-EfficientNet Diagnostic System, enabling better classification and diagnostic capabilities.

### 4.2. Proposed Methodology

A substantial collection of image features is necessary to obtain improved classification results since they immediately aid in differentiating the various picture data groups. The recall rate of approaches can be increased by employing dense DL networks to calculate a set of more efficient attributes. Since deploying these convolutional neural network (CNN) approaches depends significantly on the availability of processing power and memory requirements, there is a computational limitation on the models when deep networks are used. As a result, there are always tradeoffs between the computation cost and the evaluation’s findings. Determining a method for identifying eye illnesses that may display increased classification accuracy while maintaining computational expenses is required. This paper presents a straightforward and trustworthy computational approach to enhance the model’s ability to classify different abnormalities.

Originally, the Mobile inverted bottleneck (MBConv) block is a foundational unit of the MobileNetV2 network, establishing a methodical progression of operations designed for efficient CNN computation, particularly suited for mobile or edge devices due to its lightweight nature. MBConv incorporates a three-step operation as shown in Figure 4. Variations of MBConv include Fused-MBConv, which amalgamates the first pointwise convolution and the depth-wise convolution, offering computational efficiency on certain devices despite an increase in parameters. The FMBConv Block substitutes the 1 × 1 pointwise convolution with a group convolution, and the MobilenetV3-MBConv module introduces squeeze-and-excitation (SE) connections to the conventional MBConv as shown in Figure 4, providing channel-wise recalibration and potentially improving representational capabilities. However in this paper, these blocks are integrated into EfficientNet architecture.

The EfficientNet model, an upgraded version of EfficientNet V2-B4, is provided to detect HR and DR eye disorders. A further developed version of EfficientNet is EfficientNetV2 [31]. The updated EfficientNetV2 model is essentially offered to expand the resources available while retaining a high recall rate. The improved EfficientNetV2 model was developed using a composite scaling strategy that rapidly and efficiently scales a regular ConvNet while preserving its capabilities. This strategy allows the model to adjust to various resource limitations without sacrificing performance. The composite scaling approach enables the selection of the optimal network architecture, including the number of layers and the feature vector size, based on specific requirements and constraints. This flexibility ensures that the model can be adapted to different computational resources and cost considerations.

**Table 4 diagnostics-13-03236-t004:** Utilized datasets to test and compare the performance of proposed system.

Cited	Dataset Name	Images	Type	URL
[32]	Diabetic Retinopathy Detection (Kaggle Dataset)	120	DR	https://www.kaggle.com/c/diabetic-retinopathy-detection/data (accessed on 1 February 2023.)
[33]	IDRiD (Indian Diabetic Retinopathy Image Dataset)	480	DR	https://idrid.grand-challenge.org/ (accessed on 4 February 2023)
[34]	MESSIDOR (Messidor Diabetic Retinopathy Dataset)	300	DR	http://www.adcis.net/en/third-party/messidor/ (accessed on 6 February 2023)
[35]	e-ophtha (Electronic Ophthalmology Database)	100 (50 + 50)	DR and HR	https://www.adcis.net/en/third-party/e-ophtha/ (accessed on 2 March 2023)
[36]	HRF (Hypertensive Retinopathy Dataset	100	HR	https://www5.cs.fau.de/research/data/fundus-images/ (accessed on 1 March 2023)
[37]	EYEPACS	300 (150 + 150)	DR and HR	https://www.kaggle.com/c/diabetic-retinopathy-detection/ (accessed on 3 March 2023)
	Combination	6000 of Normal, 700 of HR and 700 of DR retinographs		

By employing the composite scaling strategy, the improved EfficientNetV2 model provides an optimal solution regarding network architecture and cost estimation. It allows for efficient resource utilization while maintaining the model’s capabilities and achieving high performance on the given task.

The EfficientNetV2 technique reliably completes classification operations while using fewer model parameters. Additionally, it outperforms other strategies in terms of effectiveness, like GoogleNet, AlexNet, DenseNet, ResNet, and MobileNet. Since it is a lightweight, efficient method that requires less training time and fewer parameters, EfficientNetV2 with dense layers was created for identifying HR and DR eye-related disorders. Using neural architecture search, the EfficientNetV2 method improves classification accuracy while minimizing feature vector size and training time. In addition, the EfficientNetV2 design optimizes the operational power with fused-MBConv (FMBConv) blocks, while the traditional EfficientNet approach solely uses mobile inverted bottleneck (MBConv) blocks for its depth-wise convolutions.

Although depth-wise convolutions need fewer operations, they do not fully take advantage of modern hardware accelerators. Both the MBConv and FMBConv blocks are entirely utilized by the EfficientNetV2 method to boost computation. The FMBConv uses traditional convolution layers in place of the depth-wise (3 × 3) convolution. The primary goal is to accelerate the model’s implementation while maintaining the classification outcomes displayed in Figure 4.

In order to tackle the classification challenge, we combined the B4 architecture with EfficientNetV2. The B4 base was chosen for its favorable balance between model classification performance and computational complexity. For a detailed description of the enhanced EfficientNetV2 model, refer to Table 5. The upgraded EfficientNet-V2 model incorporates several components at an advanced level. It utilizes MBConv blocks with 3 × 3 and 5 × 5 convolutions, squeeze-and-excitation blocks (SEB), and swish activation. At the bottom layers, FMB-Conv blocks are employed. These MBConv blocks maintain a beneficial residual connection across the SEB, enhancing the reliability of the classification results.

By combining the strengths of the B4 architecture with the improved features of EfficientNetV2, we aim to achieve a highly performant model for the classification task. The specific design choices, such as the selection of MBConv blocks, SEB, and swish activation, contribute to the model’s ability to capture and extract meaningful features from the input data. The swish activation function (*SAF*) has replaced the ReLU activation function (ReLUAF) in the framework because ReLU eliminates values below zero and loses a crucial aspect of the ECG signal. To determine the *SAF* (1), apply the equation shown below:(1)$$SAF\left(x\right)=X.\;Sigmoid(x)$$

FMBConv blocks were employed. Reduced model parameters and the issue of overfitting were tackled by adding a global average pooling layer after the MBConv. We also added two additional inner-dense layers on top of the ReLUAF and dropout layers, which, when displayed correctly, help in the computation of a more efficient set of image characteristics. An arbitrary dropout rate of 30% was selected to enhance the performance of the model. Finally, a Softmax layer was applied to categorize eye-related diseases.

At the outset of the framework, a Batch normalization layer was also added to down-sample the input picture sizes. Information on the layers and blocks was utilized in the proposed model as were a large amount of data and forecast outcomes. The *LF* is iteratively adjusted to minimize the error and achieve a robust fitting value.

In the specific case of the HDR-EfficientNet model, the final layer was removed, and an output neuron was added to enable the classification task of distinguishing between high-quality and distorted data. This modification allows the model to generate a prediction based on the learned features and make a binary classification decision regarding the input data’s quality. By customizing the model architecture and incorporating the output neuron, the EfficientPNet model can be tailored to perform the specific classification assignment required in the given context. The adjustment of the final layer and the addition of the output neuron contribute to the model’s ability to accurately classify the data as either high quality or distorted. As a result, an empirical methodology was used to propose the framework’s hyperparameters. In the model training phase of our suggested methodology, we used the Adadelta optimizer and a learning score of 0.1. Moreover, we trained the model for a period of 20 epochs. The cross-entropy *LF* uses the Softmax function to measure the variance between calculated and real values when performing classification tasks. The following formula is used to obtain the cross-entropy *LF*:(2)$$LF=\frac{1}{N}\;\sum_{k=1}^{n}log(\frac{e^{s_{j}}}{\sum_{i}e^{s_{k}}})$$

In this case, *N* stands for the total number of neurons, $s_{k}$ for the input vector, and $s_{j}$ for the estimated label. Just 20% of the total framework parameters may be adjusted in the model without changing the other 80%. In order to ensure that model overfitting problems were avoided, a validation set was used. The learning rate value was calculated for each parameter using adaptive moment estimation. Equations (3) and (4), respectively, illustrate how this method stores the exponential decline of the prior gradient using the im-pulse approach.
(3)$$LM_{t}=b1\;M_{t}-1+\left(1-b1\right)G_{t}$$
(4)$$V_{t}=b2\;V_{t}-1+\left(1-b2\right){G^{2}}_{t}$$

The first-moment ($M_{t}$), second-moment ($V_{t}$) vectors, constants (*b*1 and *b*2), gradient scores (*G*), and bias correction factors are parameters used in Equations (3) and (4). Mt (first-moment vector): This likely refers to the first-moment estimate of the gradient. In the context of optimization algorithms like Adam, this represents an exponentially moving average of the gradient. It is used to keep track of the average gradient over time. Vt (second-moment vector): This is the second-moment estimate of the gradient. Like the first-moment vector, it is an exponentially moving average, but this time of the squared gradient values. It is used to keep track of how much the gradients vary or change over time. *b*1 and *b*2 (constants): These constants, usually denoted as beta1 (*b*1) and beta2 (*b*2), are hyperparameters of the Adam optimizer. They control the decay rates of the first- and second-moment estimates, respectively. Typically, *b*1 is set to a value like 0.9, and *b*2 is set to a value like 0.999. G (gradient score): This likely refers to the current gradient of the parameters being optimized. The gradient indicates the direction and magnitude of the steepest ascent (for maximizing) or descent (for minimizing) of the objective function. In optimization algorithms, the gradient is used to adjust the parameters to minimize the loss function. Bias correction factors: The bias correction factors are used to account for the initializations of $M_{t}$ and $V_{t}$) at zero. In the early iterations of the optimization process, when the estimates are close to zero, it might cause biases in the calculations. The bias correction factors help in adjusting the estimates to compensate for this initialization bias. The equations you mentioned (Equations (5) and (6)) are likely used to compute the bias-corrected first-moment vector ($M_{t}$) and second-moment vector ($V_{t}$) values. These corrected values are then used to update the parameters of the model to minimize the loss function efficiently. The bias-corrected $M_{t}$, as described in Equations (5) and (6), can be used to eliminate these biases:(5)$$M_{t}=M_{t}-\left(bt1\right)$$
(6)$$V_{t}=V_{t}-\left(bt2\right)$$

Equation (7) is used in our model’s optimization strategy to update the gradient value.
(7)$$W_{t+1}=W_{t}-\eta /{(V_{t}+\epsilon \;M)\;}^{0.5\;}t$$

Here, is a constant *ϵ*, *η* is a learning rate with a score of 0.00001 used to keep the denominator from becoming zero, and *W*(*t* + 1) displays the framework parameters at a particular time (*t* + 1). Models use the loss function (*LF*) as a job to evaluate their effectiveness. Networks employ automatic learning to detect patterns in massive. The overall steps of proposed HDR-EfficientNet model is described in Algorithm 1.
**Algorithm 1:** HDR-EfficientNet model for prediction of HR and DR eye-related diseases.Step 1Input retinographics (300 × 300) images, with training labelsStep 2Output Class (HR, DR and Normal) categories of Retinograph images, labels of each class sample, and development of HDR-EfficientNet based on improved EfficientNet-V2 architectureStep 3Apply Pre-processing (X) and data-augmentation (X)Step 4[Training classifier phase]
(a)EfficientNetV2() and Evaluate-framework() to measure the main features of EfficientNet-V2 and perform model training
(b)For each data sample I in Training images do   Computer features based on Enhanced-Efficient-V2 model[end for loop]Step 5Used training images for HDR-EfficientNet, and calculate timeStep 6Identify Label = Predict (HR, DR, normal)Step 7Evaluate-Framework (Enhanced-Efficient-V2, Localize)Step 8[Testing phase]
For each tm data sample test in Testing images do(a) F = Extract-features by HDR-EfficientNet()(b) calculate confidence score, class label = Classify I(c) display class label[end for loop]Step 9[Exit algorithm]

## 5. Experimental Results

In this section, we provide a description of the experimental setup for all tests, including a concise explanation of the performance metrics used. Furthermore, we showcase the outcomes of the conducted experiments and offer an analysis of these results.

### 5.1. Experimental Setup

To assess the effectiveness of the recommended deep learning-based HDR-EfficientNet technique. Each dataset is split into a training set and a testing set using a 3:1 split, which means that in each experimental session, three-quarters of the data are used for training and the remaining one-fourth are used for testing. These retinal images were created using three different internet sources, one private source, and one. These retinograph images are resized to (300 × 300) pixels. We propose VGG16 and deep CNN deep learning techniques are used to create the system. Figure 1 displays digital fundus benchmarks using several imaging modalities.

The HDR-EfficientNet system was developed on a computer system equipped with an Intel Core i7 CPU, 16 GB of RAM, and a 4 GB NVIDIA GPU. Running on Windows 10 Professional 64-bit edition, the system utilizes TensorFlow (version 2.7) and Keras, two popular deep learning libraries.

In building and training the CNN architecture, careful consideration is given to the selection of kernel dimensions for generating feature maps from the previous stage. The convolutional layer’s weight values are adjusted accordingly, with a preference for kernel sizes of either (3 × 3) or (5 × 5). The convolutional layers employ varied window widths and values derived from the excitation objective function of each feature map during the convolution process. Similar techniques are applied to both the pooling layer and the convolutional layer. However, there is one distinction: the pooling layer utilizes a window size of (2 × 2) and sliding increments of 2 to maximize the features obtained from the previous layer. This step reduces the convolutional weights while improving the overall speed of the network. The result of the average pooling operation is then passed on to a fully connected layer for further processing.

### 5.2. Data Augmentation and Class Imbalance

We used pictures of eye-related conditions from the DR and HR datasets to train, validate, and test the proposed DL model. Images of DR, HR, and typical retinograph circumstances were included in the collection. Each image in the group has a resolution of 300 by 300 pixels. The photographs of wholesome HR and DR eye-related diseases showed the leaves in their typical, healthy form. The early and late blight photographs contrasted the two stages of the shattering eye-related disorders. We utilized the indices 0, 1, and 2 for the three classes in the dataset. The total number of photos is broken down into various categories in Table 2. The dataset has fewer photographs of healthy retinographs than the other two types of eye-related diseases. The pictures in the dataset were randomly selected to yield an 80/20 training and test set.

We added healthier eye-related photos to balance the dataset by randomly choosing ten ordinary photographs and making ten copies of each. This procedure was applied five times to balance the dataset regarding standard retinographics. The dataset displays the overall number of photos in each class after hovering. The 6000 photographs of traditional ideas, and the 1400 photographs of HR and DR were initially included in each category. After data augmentation, each type contained 12,000 photos of regular, HR, and DR. In total, we used 36,000 retinograph images to test the performance of the proposed system.

We increased the training set and normalized the data to train the model without risking overfitting. The photos underwent transformations for enhancement based on the parameters listed in Table 6. Figure 5 is a diagrammatic depiction of the augmentation process.

### 5.3. Model Evaluation Metrics

In this study, a specific dataset was utilized for the purpose of classifying instances into nine distinct categories. To comprehensively evaluate the classification results, a range of performance metrics were employed, encompassing the confusion matrix, accuracy, precision, recall, F1 score, and the Matthews correlation coefficient (*MCC*).

The confusion matrix serves as a detailed breakdown of the predicted and actual class assignments, capturing a variety of possible outcomes. It specifically includes true positive (*TP*) and true negative (*TN*) values, signifying the instances correctly identified as belonging to the HR and DR classes, respectively. Conversely, false positive (*FP*) and false negative (*FN*) values indicate instances inaccurately classified as HR, DR, or normal, respectively. For a comprehensive assessment of the model’s performance, the accuracy, recall, and F-measure were calculated using the macro-average technique for each of the nine classification classes. The macro-average approach treats all classes equally when calculating these metrics, offering a holistic perspective across all categories. Additionally, the Matthews correlation coefficient (*MCC*) was employed, as it is a robust metric capable of delivering a balanced evaluation even when classes vary significantly in size. The *MCC* accounts for true positives, true negatives, false positives, and false negatives, allowing for a comprehensive understanding of the model’s effectiveness.

Collectively, these metrics provide a comprehensive framework for assessing the classification model’s performance and offer valuable insights into its efficacy across diverse categories. This approach ensures a thorough evaluation of the model’s capabilities and its ability to accurately classify instances within each specific category.

Following is how these metrics are calculated:(8)$$Accuracy\;\left(ACC\right)=\frac{TP+TN}{TP+TN+FP+FN}$$
(9)$$Precision\;\left(PR\right)=\frac{TP}{TP+FP}$$
(10)$$Senstivity\left(SE\right)=Recall\left(RC\right)=\frac{TP}{TP+FN}$$
(11)$$Specificity\;\left(SP\right)=\frac{TN}{TN+FP}$$
(12)$$F1score=2\times \frac{PR\times RC}{PR+RC}$$
(13)$$MCC=\frac{TP\times TN-(FP\times FN)}{\left(TP+FP\right)(TP+FN)(TN+FP)(TN+FN)}$$

### 5.4. Results Analysis

We conducted experiments using several powerful CNN models, including VGG16, AlexNet, InceptionV3, GoogleNet, Xception, MobileNet, SqueezeNet, and SqueezeNet-Light, to recognize multiclass scenarios. Additionally, we compared our results with state-of-the-art (SOTA) techniques. To assess the effectiveness of these approaches, we utilized both validation and testing splits. The proposed HDR-EfficientNet model accuracy versus loss validation curves is shown in Figure 6. The testing data were obtained from separate sources, while the validation split was created using the same sources as the training data. This differentiation allows us to evaluate the models’ performance on unseen data and verify their generalization capabilities.

We examined various scenarios, including two-class and three-class classifications. For all experiments, we employed a 10-fold cross-validation technique, where the dataset was divided into ten subsets of roughly equal size. Each subset was used as a validation set once, while the remaining nine subsets were used for training. This approach helped to ensure a robust evaluation by considering different combinations of training and validation data. To assess the models’ efficacy, we calculated performance measures such as the accuracy, precision, recall, F1 score, and Matthew’s correlation coefficient (MCC) for each fold in the cross-validation process. The means of these metrics were then calculated and reported in the subsequent sections of this study, allowing us to evaluate the overall performance of the models across different classification scenarios. By employing these evaluation techniques, we aim to gain insights into the effectiveness of the CNN models and compare their performance in various multiclass scenarios.

With VGG16 serving as the core architecture for training, accuracy, and validation, along with a training loss and a validation loss function, we start testing our suggested HDR-EfficientNet model on the various datasets. Figure 7 demonstrates the effectiveness of our suggested HDR-EfficientNet approach. To obtain a training accuracy and validation accuracy of above 96%, the training and validation procedures only needed to be iterated a total of 10 times. In addition, we were able to obtain a loss function below 0.1 for both training and validation data, further demonstrating the efficacy of our proposed method. The confusion matrix must first be gathered to appropriately evaluate detection performance.

### 5.5. Computational Cost

HDR-EfficientNet is a family of EfficientNet models that are designed to achieve state-of-the-art performance while maintaining a high level of efficiency in terms of computational complexity. The models in the EfficientNet family are scaled versions of a base architecture, where the scaling is performed uniformly across multiple dimensions, including depth, width, and resolution. The computational complexity of EfficientNet models can be characterized by two main aspects: Floating-point operations (FLOPs) and the number of parameters, which measure the number of floating-point operations required for the model to make predictions. EfficientNet models typically have a lower number of FLOPs compared to other deep neural network architectures of similar performance. This reduction in FLOPs is achieved by carefully balancing the model’s depth, width, and resolution during the scaling process.

Number of parameters: The number of parameters in a model reflects its memory requirements and affects both the model’s training time and inference time. EfficientNet models strike a balance between model size and performance by scaling the number of parameters appropriately based on the desired level of efficiency. EfficientNet models achieve a good trade-off between computational complexity and performance by employing a compound scaling method that optimizes the depth, width, and resolution simultaneously. This allows EfficientNet models to provide high accuracy while being computationally efficient, making them suitable for a wide range of applications, especially in resource-constrained environments such as mobile devices or edge computing platforms.

Let us consider HDR-EfficientNet, which is the smallest and least computationally complex variant of the EfficientNet family. The FLOPs and number of parameters for HDR-EfficientNet are as follows:

FLOPs: Approximately 0.39 billion (390 million) FLOPs. This indicates the number of floating-point operations required to process a single input image through the network.

Number of parameters: Approximately 5.3 million parameters. This represents the number of learnable weights in the model, which are adjusted during the training process. Compared to larger models such as EfficientNet-B7, which have significantly higher FLOPs and parameters, HDR-EfficientNet offers a less computationally intensive option while still achieving reasonable performance.

It is important to note that these numbers are approximate and can vary depending on the specific implementation and framework used. Additionally, the computational requirements can vary across different layers and operations within the network. HDR-EfficientNet models are designed to provide an efficient balance between computational complexity and performance, allowing for effective deployment on a range of devices and platforms. The computational parameters are mentioned in Table 7.

One of the advantages of HDR-EfficientNet is its ability to reduce the number of parameters in the network, leading to a smaller model size. This reduction in parameters was achieved in part by using separable transfer networks. Table 8 provides a comparison of the parameter count in the convolutional layers of various CNN models, including VGG16, AlexNet, InceptionV3, GoogleNet, Xception, MobileNet, SqueezeNet, and the proposed HDR-EfficientNet. The results revealed that the suggested model significantly reduced the number of parameters in the convolutional layer. Importantly, the reduction in parameters did not result in degenerate models but rather improved network generalization. In other words, the suggested model maintained its performance while being more efficient in terms of parameter utilization. In conclusion, the suggested HDR-EfficientNet model demonstrates superior performance, particularly when applied to large datasets, and establishes a solid foundation for its utilization in differentiation between HR and DR eye-related diseases. Furthermore, its faster running speed compared to certain other conventional models makes it a promising choice for real-time applications.

### 5.6. Performance Analysis

In the hyperparameter tuning process for the HDR-EfficientNet system, we aimed to optimize the learning rate, a crucial parameter affecting the training of the model. Using a grid search approach, we examined a range of learning rates, such as 0.001, 0.01, and 0.1. For each learning rate and batch size, we built the HDR-EfficientNet architecture, trained it on retinograph images, and evaluated its performance on a validation dataset. The learning rate that resulted in the highest validation accuracy was chosen as the optimal one. This process ensures that the model converges effectively during training, leading to better classification results.

In the context of performance analysis, the selection of dropout values and initial learning rates for the EfficientNet model during training was carried out randomly to mitigate the impact of manual parameter tuning. To reduce the influence of human bias, the network was trained to autonomously determine the optimal EfficientNet architecture. The dropout rate varied from 0.2 to 0.6, and the model incorporated batch normalization layers, enabling the use of higher initial learning rates ranging from 102 to 104. Training was performed using the Adam optimization technique with random choices of initial learning rates and dropout values under batch sizes of 32, 64, and 128. Each network for diverse training batches encompassed 30 distinct initial sequence parameters. A maximum of 10,000 training sessions was allowed per network with combined parameters, but training ceased if the verification set loss reached a plateau within ten sessions. To evaluate the system’s performance, various metrics including accuracy (ACC), specificity (SP), sensitivity (SE), precision (PR), recall (RL), F1-score, and Matthews correlation coefficient (MCC) were computed using statistical analysis. These metrics facilitated performance assessment and comparison with pre-trained transfer learning algorithms. The assessment spanned multiple experiments that gauged accuracy across different convolutional layers of various models such as VGG16, AlexNet, InceptionV3, GoogleNet, Xception, MobileNet, SqueezeNet, and the proposed HDR-EfficientNet model. Additionally, the area under the receiver operating curve (AUC) was used to demonstrate the efficacy of the training and validation datasets through 10-fold cross-validation tests. Figure 6, Figure 7 and Figure 8 visually present the optimal plot loss, accuracy, AUC, and recall achieved during training and validation with data augmentation, conducted for 40 epochs, specifically for the HDR-EfficientNet model.

The model’s label for each category remains unmuddied even with a limited dataset training sample. The two groups have been categorized accurately. Figure 6 shows that there is less uncertainty when using our recommended model, HDR-EfficientNet, for detection. The training phase is complete in multiple CNN and VGG19 architectures with eight to sixteen stages and pre-processing to improve contrast to compare the retinal fundus datasets. Table 8 provides an illustration of the results. It is crucial to remember that the same number of epochs were used to train each of the CNN and VGG16 deep learning models. By employing the top-performing network to train an identical classic convolutional network, we obtained a 59% improvement in validation accuracy. The sensitivity, specificity, accuracy, and area under the curve (AUC) measurements were used to evaluate the proposed CAD performance in comparison to that of standard CNN, DRL, trained-CNN, and trained-DRL models. For the trained-CNN models on this dataset, the average values for SE, SP, ACC, and AUC were 81.5%, 83.2%, 81.5%, and 0.85, respectively. In contrast, the SE, SP, ACC, and AUC metrics values for the HDR-EfficientNet model were 94%, 96%, 95%, and 0.96, respectively. By combining the abilities of HDR-EfficientNet on four annotated fundus sets that are not prone to overfitting issues, the developed DR and HR system produced results as shown in Table 9 that were superior to those of deep-learning models.

**Table 8 diagnostics-13-03236-t008:** Metrics are used to evaluate the HDR system’s performance.

Retinopathy Type	^1^ SE	^2^ SP	^3^ ACC	^4^ AUC	^5^ MCC
Diabetic Hypertension (HR)	93%	96%	94%	0.95	0.76
Diabetic Retinopathy (DR)	95%	96%	95%	0.96	0.67
Normal	93%	96%	94%	0.95	0.76
Average Result	94%	96%	95%	0.96	0.60

^1^ Sensitivity, ^2^ Specificity, ^3^ Accuracy, ^4^ Area under the curve, ^5^ Matthews correlation coefficient.

**Figure 8 diagnostics-13-03236-f008:**
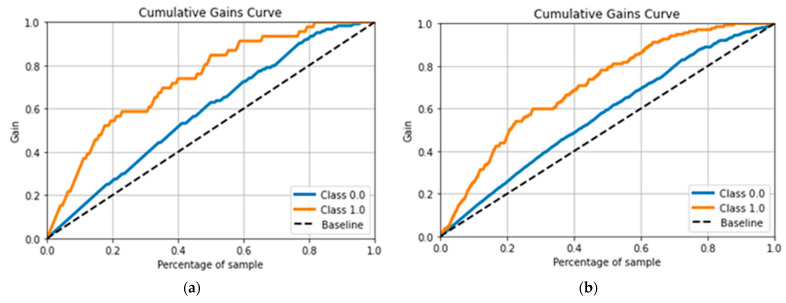
Plot of AUC where class 0 represents the hypertension and class 1 shows the diabetes without data augmentations with 10-fold cross validation. Whereas the figure (**a**) shows the dataset obtained from (Kaggle-Dataset [32], IDRiD [33], MESSIDOR [34]), and (**b**) shows the datasets obtained from (Kaggle-Dataset [32], IDRiD [33], MESSIDOR [34], e-ophtha [35], HRF [36], and EYEPACS [37]). In this experiment, we used 600 retinograph images of equal size of each category (DR, HR and normal).

**Experiment 1:** In this section of the paper, we compare the proposed work to previous DL approaches and show how its superiority over those approaches may be demonstrated. A number of well-known deep learning frameworks, such as VGG16 [20], VGG19 [21], MobileNet [22], ResNet50 [23], and DenseNet-101 [24], were proposed for this purpose. We were able to evaluate various DL architectures from the perspectives of model structure and performance by comparing the total number of model parameters and accuracy. The evaluation’s findings are displayed in Table 9 of the report. The numbers show that, in comparison to previous DL frameworks, our technique is both successful and efficient. Apparently, this study has the fewest model parameters—11 million. The VGG19 model is more expensive in terms of model structure with a total of 1.96 million parameters. ResNet50 achieved the worst performance result in terms of model correctness, scoring 73.75%. MobileNet has the second-worst performance rating with a score of 78.84%. With an accuracy score of 93.93%, the DenseNet technique performs better, but its 40 million parameters and intricate network topology make it difficult to implement. In contrast, our method works well, scoring 98.12% accurate and using 11 million model parameters. Obviously, our model’s score is 98.12%, whereas the average for similar techniques is 83.92%. As a result, we have seen an increase in performance of 14.20%, which amply demonstrates the effectiveness of our strategy.

**Experiment 2:** In our evaluation, we adopted a robust 10-fold cross-validation testing strategy to comprehensively compare the statistical metrics obtained from our developed HDR-EfficientNet model with various other transfer learning (TL) algorithms. These included established models like VGG16, AlexNet, InceptionV3, GoogleNet, Xception, MobileNet, SqueezeNet, and our own SqueezeNet-Light. We assessed the classification outcomes of these pretrained deep learning (DL) models across different batch sizes, namely 16, 32, and 64, and presented the results in Table 10, Figure 9, and Table 11, respectively. Interestingly, the performance of our HDR-EfficientNet model remained consistent across all batch sizes, with an identical classification outcome. However, the impact of batch size was evident in terms of the number of model parameters and computational time. This finding contrasted with the other pre-trained TL algorithms, where the classification results held steady irrespective of batch size. Our developed approach exhibited impressive results, boasting a sensitivity (SE) of 94%, specificity (SP) of 96%, accuracy (ACC) of 95.6%, precision (PR) of 94.12%, F1-score of 95.2, and Matthews correlation coefficient (MCC) of 96.7. Notably, the system also demonstrated low training error (0.76) in accurately identifying various classes of eye-related diseases within a multiclass framework. These findings underscore the efficacy and robustness of our approach in classification tasks pertaining to eye-related diseases.

The other techniques’ fairly complicated model structures, which lead to problems with model overfitting, are mostly to blame for these higher model classification outcomes. Comparatively, our strategy is more flexible and better equipped to address the overfitting problem. Additionally, our approach incorporates layers at the end of the network structure and leverages pixel and channel attention throughout the feature computation step, which aids in better recognizing the meaningful set of image characteristics and improves the cataloguing score. Therefore, it can be stated that we have offered a system that is both efficient and effective for memorizing the several categories of illnesses impacting HR and DR eye-related sickness.

**Experiment 3:** In this experiment, we focused on investigating the influence of different optimization techniques on classification performance. To build an efficient SqueezeNet-Light model, we employed various optimizers. Among these, adaptive algorithms like Adam exhibit rapid convergence, whereas stochastic gradient descent (SGD) optimizers demonstrate better generalization, particularly when presented with new data. In an effort to merge the strengths of both optimizer types, AdaBelief was previously developed to manage the loss function. AdaBelief is specifically designed to handle cases characterized by “large gradient, small curvature,” an aspect that Adam might struggle with.

Within the context of the same 10-fold training data setup, Figure 10 offers a comparative analysis of optimizers, including the weighted variant. We compared the AdaBelief optimizer, which incorporates the learning rate, weight decay, and momentum settings of 1 × 10^−5^, 1 × 10^−8^, and 0.9, respectively, with other optimization methods. When utilizing the AdaBelief optimizer with momentum, we extended the epoch number to 40. This decision was informed by earlier experiments that demonstrated the loss function’s value continuing to decrease beyond the 30th epoch, implying a lack of convergence by the 30th epoch’s conclusion. It is important to note that for the purposes of this study, we maintained a consistent epoch number of 40 across all optimizers to ensure a fair comparison. Table 8 consolidates the numeric outcomes of these experiments. Notably, when the AdaBelief optimizer was employed, the sensitivity (SE) value exhibited a notable increase, reaching 94%. Consequently, based on these findings, we identify the AdaBelief optimizer as the optimal choice for the optimization method in these experiments.

**Experiment 4:** In this particular experiment, we delved into the evaluation of various loss functions and their impact on classification performance. The outcomes highlighted the efficacy of the weighted-cross entropy loss function in enhancing classification accuracy by addressing class imbalance issues. Notably, when compared to the regular cross entropy loss, the weighted-cross entropy loss consistently yielded better results. Specifically, for different metrics, the cross-entropy loss values were 82.2 (SE), 97.8 (SP), 96.9 (ACC), 76 (PR), 79 (F1-score), and 82.2 (MCC), while the corresponding values for the weighted-cross entropy loss were significantly improved at 94 (SE), 96 (SP), 95.6 (ACC), 94.12 (PR), 95.2 (F1-score), and 96.7 (MCC). Based on these results, it becomes evident that the weighted-cross entropy loss function is more effective in enhancing classification performance. For a comprehensive overview of these outcomes, Figure 11 presents the detailed results derived from this experiment, showcasing the superior performance achieved through the use of the weighted-cross entropy loss function.

Figure 12 presents a visual representation of the outcomes achieved through our proposed HDR-EfficientNet classifier, specifically in the context of identifying healthy retina (HR) and diabetic retinopathy (DR). While the testing splits for other disease classes were collected from diverse sources, our focus was on assessing various pre-trained transfer learning (TL) convolutional neural network (CNN) architectures. The architectures evaluated included VGG16, AlexNet, InceptionV3, GoogleNet, Xception, MobileNet, and SqueezeNet techniques. This comprehensive evaluation aimed to distinguish between different eye-related diseases across various scenarios. In order to circumvent the challenge of requiring extensive labeled data for CNN architectures, we employed data augmentation techniques. The findings of our experiments revealed that our proposed model outperformed these CNN architectures in the task of disease classification. Notably, our suggested HDR-EfficientNet architecture achieved an accuracy rate of 95.6% in effectively recognizing and classifying various types of retinograph images within a multiclass framework. This result highlights the potency of our approach and the HDR-EfficientNet model in addressing the complexities of eye-related disease identification.

### 5.7. Visualization of Features Detected by HDR-EfficientNet

Gradient-weighted class activation mapping (Grad-CAM) is a technique used in DL to visualize and understand the regions of an image that a model focuses on when making a prediction. It helps identify which parts of the image contribute most to the network decision. By applying Grad-CAM to retinograph images for both DR and HR, you can identify the specific regions that the model focuses on when diagnosing each eye-related disease. This can provide valuable insights into the patterns and features that the model uses for classification. Keep in mind that Grad-CAM visualizations are not a replacement for clinical expertise. They provide an additional tool for understanding how an HDR-EfficientNet model arrives at its predictions.

Apply Grad-CAM to find patterns of diabetic retinopathy (DR) and hypertensive retinopathy (HR), we have to perform the different steps. First, choose HDR-EfficientNet DL model as a pretrained architecture on a large dataset. Next, we must remove the final classification layer of the pretrained model and replace it with a global average pooling layer. This allows you to generate a heatmap for the entire image. Afterwards, for each image, feed it through the modified model and compute the gradients of the predicted class score with respect to the feature maps in the last layer. Overlay the normalized heatmap on the original retinograph image to highlight the regions that the model used to make its prediction. Finally, we must display the retinograph image with the overlay heatmap to visualize the areas that contributed most to the classification decision as shown in Figure 13.

## 6. Discussions

The implementation of HDR-EfficientNet for classifying HR (hypertensive retinopathy) and DR (diabetic retinopathy) involved using VGG16 as a pretrained model and a trained CNN model as input. The goal was to construct a hierarchical structure that facilitates learning specialized features without the need for complex feature selection or image-processing techniques. The architecture employed deep learning techniques to automatically extract information from input images without human intervention. HDR-EfficientNet was integrated with VGG16 and convolutional blocks to create an enhanced architecture that generates more generalizable features.

Using a scratch-based training approach, the VGG16 layer was adapted to acquire localized and trained features from four HR-related lesions. The CNN model consists of convolutional, pooling, and fully connected layers, which must be qualified and validated to effectively capture relevant data for building the model. These layers may not be optimal for recognizing HR in retinograph images. Deep residual connections were incorporated to generate highly specialized features to address this, moving away from feature-based categorization algorithms that require human input.

Previous attempts to categorize ocular problems, including those related to HR, have employed deep learning techniques instead of traditional machine learning approaches, as discussed in Section 3. Conventional methods faced significant challenges when developing automated HR systems. One issue is the reliance on complex pre- or post-image processing techniques to identify and extract clinical features associated with HR, as there are no datasets with expert annotations characterizing HR-related lesion patterns. This lack of annotated data makes it difficult for automated systems to recognize specific disease symptoms. Researchers have combined classic and state-of-the-art deep learning techniques, utilizing manually generated characteristics to train the network. However, there is a need for an automated method to determine the most essential features.

Deep-learning models have demonstrated superior results to traditional methods, despite challenges in accurately communicating decision-making weights to deeper network levels. While all models used the same weighting method, some models automatically learned features by training from scratch. This poses a challenge in effectively conveying decision-making weights to lower layers of the network.

The effective identification of HR was made possible thanks to an independent feature learning technique. However, the handcrafted-based classification methods for identifying HR diseases need computationally costly algorithms to pre-process, segment, and localize HR-related data. The necessary components were extracted carefully; other crucial signs, such as identifying cotton wool spots or hemorrhages, were not. Instead of focusing on image processing algorithms, the HDR-EfficientNet system was developed in this work to address the issues by classifying images into HR and non-HR using two multi-layer deep learning techniques. Below, we list the HDR-EfficientNet system’s main accomplishments. This study used residual blocks and a convolutional neural network (CNN) to construct two new deep-learning techniques. The hierarchy of characteristics was built by training the initial CNN model on four different HR lesions. Finding the feature maps with the most relevant data was performed using the second residual block, which increased the efficiency of the learning process. It is based on a color space that is perceptually oriented and is the first HR categorization system created in this study. The deep features are categorized using a Softmax classifier and the HDR-EfficientNet model.

This is the first attempt at an automated HR sickness detection system that we are aware of. To achieve more feature generalization, the multilayer deep learning network must be trained on various samples while creating the HDR-EfficientNet system discussed in this article. Automatic feature learning was accomplished using an HDR-EfficientNet with three residual blocks and a revolutionary deep learning-based approach. The recommended HDR-EfficientNet approach; however, erroneously classifies a few data points. There is a graphic explanation in Figure 2. It was a severe example of hypertensive retinopathy (HR), and we will talk more about this topic in future research. In terms of HR recognition accuracy, the HDR system outperformed the most recent systems, RBM-DNN [25], CNN-technique [26], and EfficientNet [31], as shown in Table 4. The HDR-EfficientNet system was developed using learned features and deep residual learning techniques.

### 6.1. Research Highlights

The effective and light deep learning (DL) method known as HDR-EfficientNet is developed in this paper. With reduced processing effort, it produces improved DR and HR eye-related classification results and is skilled at estimating important and distinguishing sample features.To create a lightweight, efficient transfer learning (TL) architecture, the model makes use of the pixel and channel attention technique throughout the feature computing phase.Transfer learning and multi-class focal loss are employed to resolve the issue of class imbalance and network overfitting, thereby increasing the accuracy of identifying HR- and DR-infected regions.We performed comprehensive comparative analyses to confirm the classification results using a range of retinal fundus images to demonstrate the effectiveness of the suggested HDR-EfficientNet model.To make the training and testing datasets larger and more evenly distributed, we used data augmentation techniques. The classifier’s generalization abilities improve with the use of various data augmentation strategies.

### 6.2. Advantages of HDR-Efficient System

The HDR-EfficientNet system offers several advantages for classifying HR (hypertensive retinopathy) and DR (diabetic retinopathy):The system leverages deep learning techniques to automatically learn and extract relevant features from retinograph images without the need for manual feature engineering. This ability to learn discriminative features directly from the data enhances accuracy and reduces the reliance on domain expertise.By combining VGG16 with convolutional blocks and deep residual connections, the system creates a hierarchical feature representation. This allows the model to capture both low-level and high-level features, enabling better discrimination between different eye-related conditions.The integration of deep residual connections and hierarchical feature learning contributes to improved generalization and robustness. The system can better differentiate between subtle variations in retinograph images associated with HR and DR, even in the presence of noise and variations.Unlike traditional approaches that heavily rely on complex and computationally intensive image processing techniques, HDR-EfficientNet focuses on direct image classification. This streamlines the process, making it computationally efficient and less prone to errors introduced by preprocessing steps.The use of pretrained models, such as VGG16, allows the system to leverage knowledge learned from large datasets and adapt it to the specific task of classifying HR and DR. This results in faster convergence during training and potentially better feature extraction.The system architecture enables the extraction of high-level abstract features that capture intricate patterns and abnormalities associated with HR and DR. This makes it capable of identifying features that might be challenging for human experts to manually define.This study suggests that the HDR-EfficientNet system outperforms existing methods in terms of accuracy for HR recognition. This superior performance indicates its potential to become a valuable tool for clinical diagnosis and decision making.By automating the feature extraction process, the HDR-EfficientNet system minimizes the need for human intervention in the classification process. This reduces subjectivity and variability that might arise when relying solely on human expertise.The HDR-EfficientNet system’s architecture can be scaled to accommodate larger datasets and extended to other similar medical image classification tasks. Its hierarchical nature allows for seamless adaptation to various complexities and quantities of data.The system’s effectiveness in classifying HR and DR suggests potential for further research and refinement. It can serve as a foundation for future studies focused on addressing specific challenges and improving overall performance.

Lastly, the HDR-EfficientNet system’s advantages lie in its ability to autonomously learn features, its hierarchical feature representation, robustness, and potential to outperform traditional approaches. These attributes make it a promising tool for accurate and efficient classification of HR and DR conditions, ultimately contributing to better clinical diagnosis and patient care.

### 6.3. Limitations of HDR-Efficient System

These limitations and future research directions highlight areas where the HDR-EfficientNet system can be improved, refined, and adapted for more effective and responsible clinical use. Table 12 outlines the limitations of the HDR-EfficientNet system for classifying HR and DR, along with potential avenues for future research:

## 7. Conclusions

Several computer techniques that can recognize hypertensive retinopathy (HR) and diabetic retinopathy (DR) from colored fundus images have been developed too far. On the other hand, contemporary methods concentrate on describing a variety of HR-related lesions and classifying them using machine learning techniques. Domain knowledge in feature selection and image processing is required to construct the DR and HR identification system. There are not many methods available right now that use deep learning (DL) models to categorize illnesses of the eye. Due to the small sample size of existing data, it was difficult to generalize these methods for use in detection of DR and HR. There is also an inaccuracy in the classification system. In this study (HR), we develop a novel multi-layer deep CNN (HDR-EfficientNet) for hypertensive and diabetic retinopathies classification by combining a features-training technique based on the VGG-16 model. To extract features from retinal fundus images and label them as having HR or not having HR, the HDR-EfficientNet method employs a multilayer architecture comprised deep residual learning blocks and trained features. By modifying the VGG16 network design, the HDR-EfficientNet system can also automatically learn and classify features. The model was created using a supervised learning technique, wherein it was fed examples of both healthy and sick leave and taught to recognize the differences between image classes. The updated EfficientNetV2 model has the potential to be more accurate and resilient because of its capacity to capture more low-level information and patterns in the images. To further boost the model’s performance, strategies including regularization, data augmentation, and transfer learning can be used. The ophthalmologist can make an informed decision because of the HDR-EfficientNet technology’s ability to identify HR. Additionally, it aids in screening sizable populations. The test findings demonstrate that the proposed system can accurately identify hypertensive retinopathy.

## Figures and Tables

**Figure 1 diagnostics-13-03236-f001:**
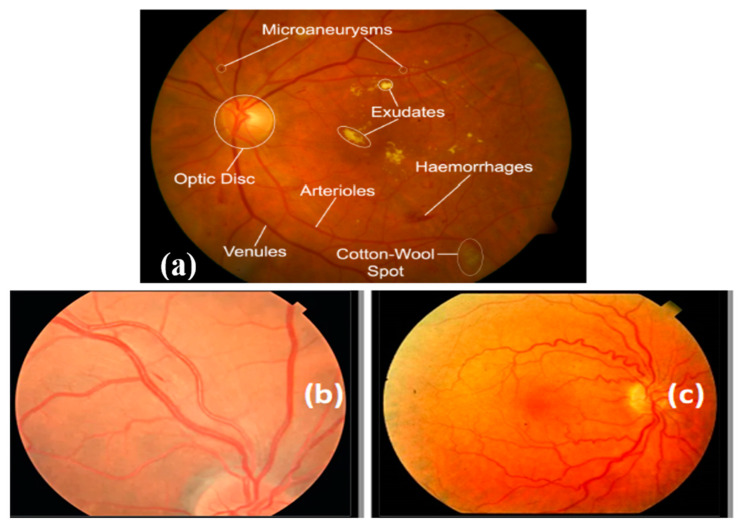
A visual example of hypertensive retinopathy (HR), where figure (**a**) shows the sample of different HR-related lesions, and figures (**b**,**c**) represents the HR images.

**Figure 2 diagnostics-13-03236-f002:**
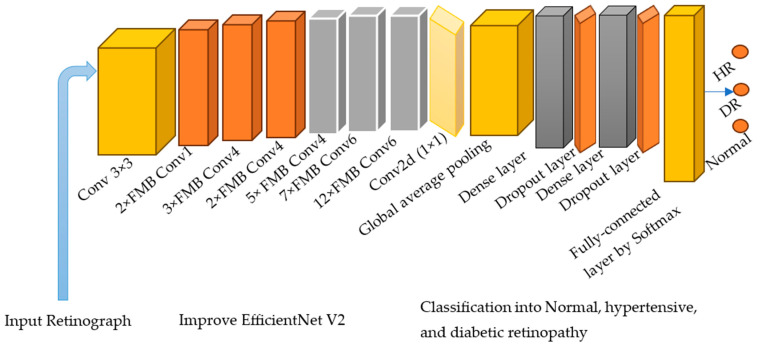
A representation of the proposed HDR-EfficientNet model’s flow for classifying eye-related disorders.

**Figure 3 diagnostics-13-03236-f003:**
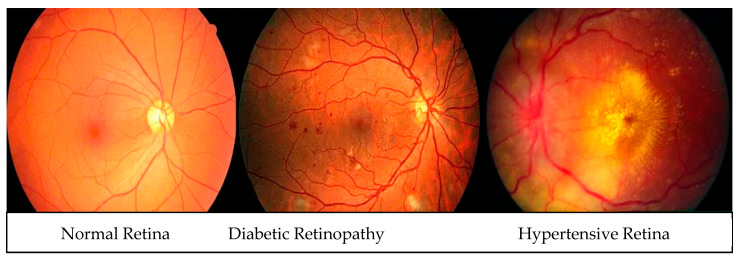
An illustration showing images of healthy or normal, diabetic, and hypertensive retinopathy (HR).

**Figure 4 diagnostics-13-03236-f004:**
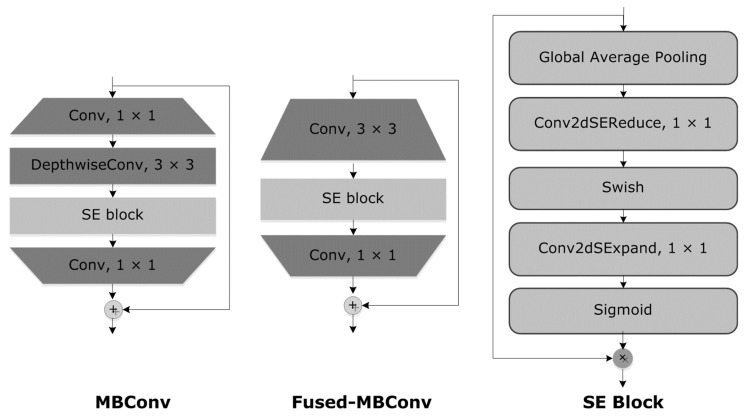
Graphic form of mobile inverted bottleneck layer (MBConv4), Fused-MBConv4, and SE block.

**Figure 5 diagnostics-13-03236-f005:**
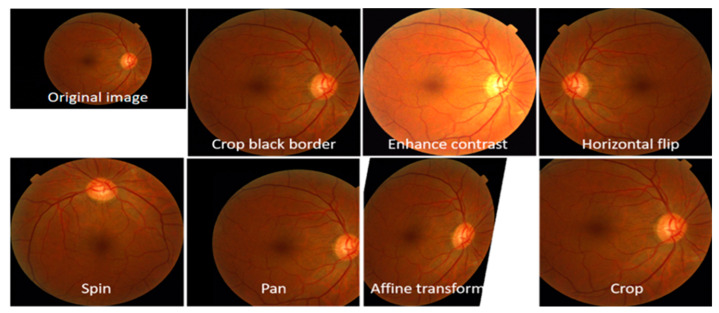
A visual illustration of the suggested data augmentation.

**Figure 6 diagnostics-13-03236-f006:**
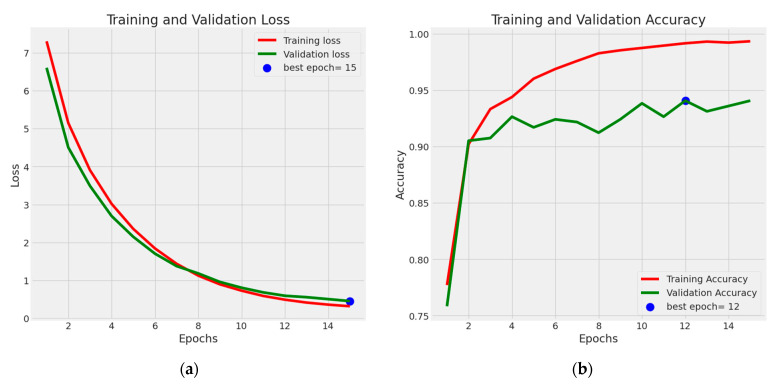
Proposed architecture accuracy versus loss on 14 number of epochs, where figure (**a**) shows the training and validation loss, and figure (**b**) shows the training and validation accuracy.

**Figure 7 diagnostics-13-03236-f007:**
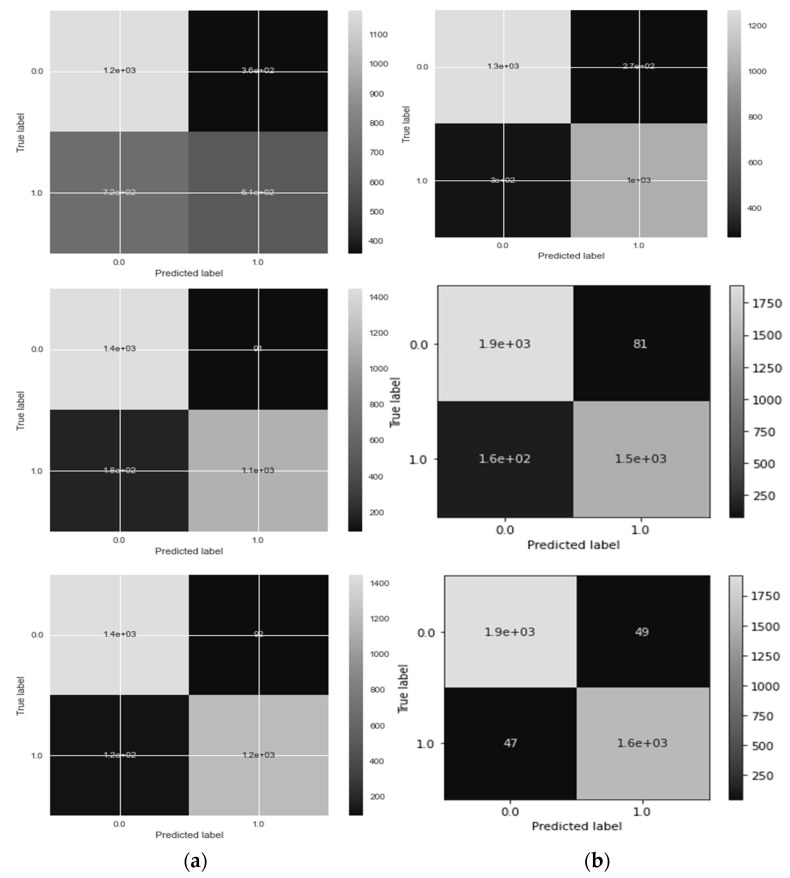
Accuracy on training and validation datasets for proposed HDR-EfficientNet system are shown visually in figure (**a**), and in figure (**b**), respectively, with respect to different split ratios (20–80%, 30–70%, 40–60% from top to bottom).

**Figure 9 diagnostics-13-03236-f009:**
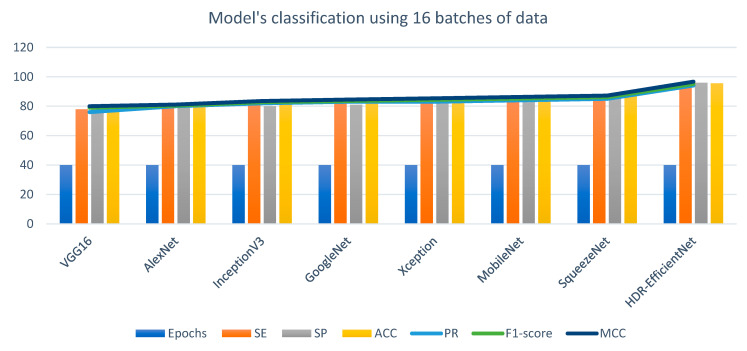
Results of the proposed system model’s classification using 16 batches of data, where SE: Sensitivity, SP: Specificity, MCC: Matthews correlation coefficient, PR: Precision, ACC: Accuracy.

**Figure 10 diagnostics-13-03236-f010:**
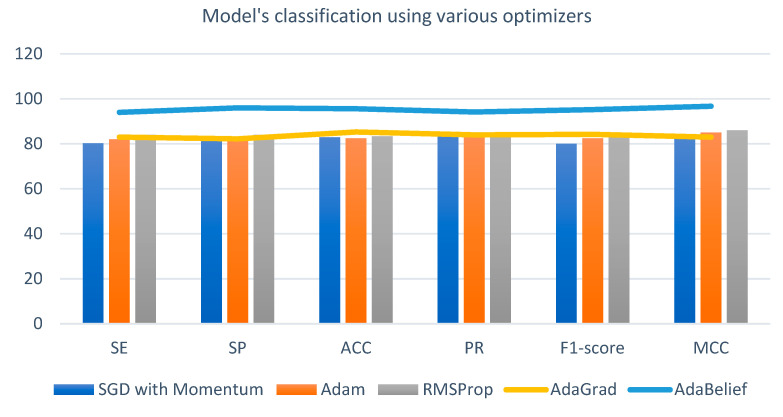
Results of the proposed system’ model’s classification using various optimizers, where SE: Sensitivity, SP: Specificity, MCC: Matthews correlation coefficient, PR: Precision, ACC: Accuracy.

**Figure 11 diagnostics-13-03236-f011:**
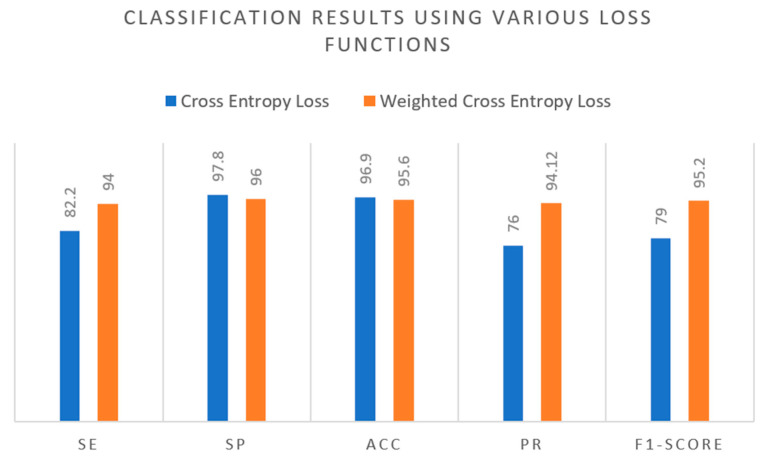
Proposed system model’s classification results using various loss functions, where SE: Sensitivity, SP: Specificity, PR: Precision, ACC: Accuracy.

**Figure 12 diagnostics-13-03236-f012:**
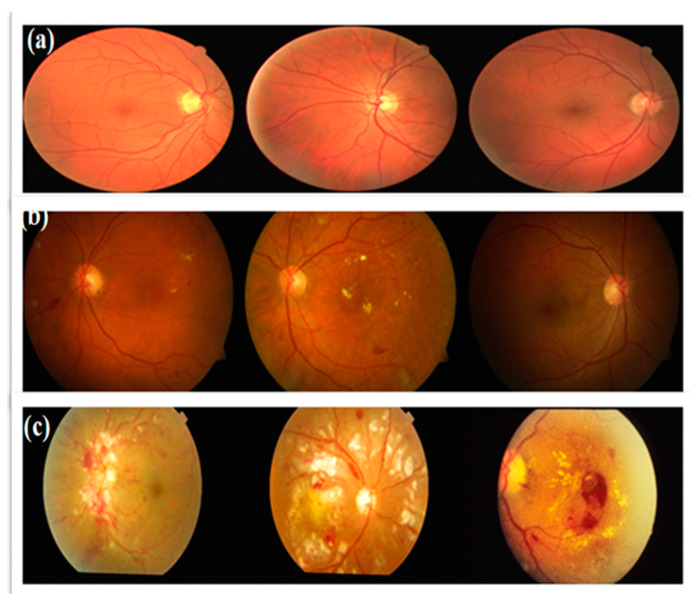
A visual example of proposed HDR-EfficientNet classification: (**a**) normal, (**b**) diabetic retinopathy, and (**c**) hypertensive retinopathy.

**Figure 13 diagnostics-13-03236-f013:**
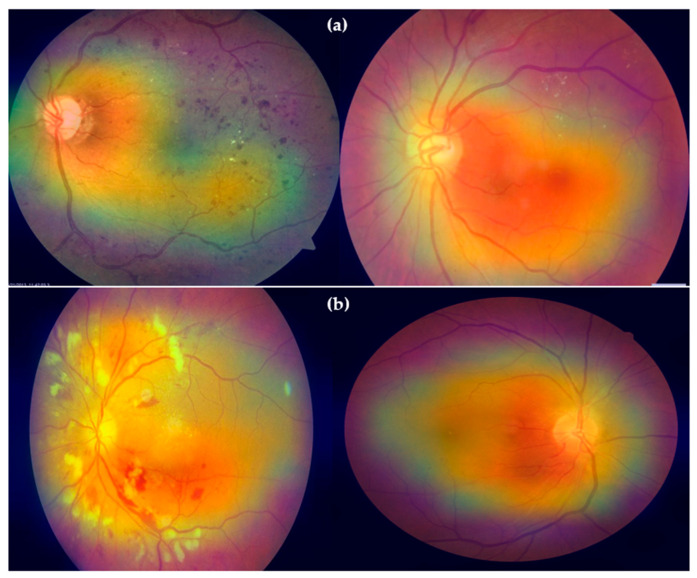
A visual example of proposed HDR-EfficientNet classification overlay heatmap by Grad-CAM approach on (**a**) diabetic retinopathy, and (**b**) hypertensive retinopathy images.

**Table 1 diagnostics-13-03236-t001:** A table summarizing the clinical importance of the HDR-EfficientNet system for classifying HR and DR.

Clinical Importance	Description
Early Disease Detection	Facilitates early identification of HR and DR, enabling timely intervention and preventing disease progression.
Reduced Subjectivity	Minimizes subjective interpretation and variability, leading to more consistent and objective diagnostic outcomes.
Enhanced Diagnostic Speed	Rapidly analyzes retinograph images, providing prompt diagnostic results for timely decision-making.
Resource Allocation	Efficiently allocates healthcare resources to patients with confirmed conditions, optimizing patient care.
Clinical Decision Support	Assists healthcare professionals by offering additional insights and information for diagnosis and treatment planning.
Telemedicine and Remote Care	Supports remote assessment of retinograph images, enabling informed decisions in telemedicine scenarios.
Reduced Workload	Automates preliminary image analysis, relieving clinicians of time-consuming tasks and focusing on patient care.
Population Screening	Aids in population-level screening programs, identifying individuals at risk and in need of further evaluation.
Access to Expertise	Provides accurate assessment in regions with limited access to specialized ophthalmologists, improving care availability.
Research and Insights	Contributes to research efforts, generating valuable data for better understanding of disease patterns and outcomes.
Patient Education	Utilizes visual outputs and results to educate patients about their condition, fostering understanding and adherence to treatment plans.

**Table 2 diagnostics-13-03236-t002:** A table summarizing the different patterns observed in retinograph images for the diagnosis of diabetic retinopathy (DR) and hypertensive retinopathy (HR).

Patterns	Diabetic Retinopathy (DR)	Hypertensive Retinopathy (HR)
Microaneurysms	Small outpouchings in blood vessels	N/A (not a common feature)
Hemorrhages	Small spots or larger blotches	Flame-shaped hemorrhages, linear bleeding
Hard Exudates	Yellow/white deposits from leaking vessels	N/A (not a common feature)
Cotton Wool Spots	Fluffy white/grey lesions with indistinct borders	N/A (not a common feature)
Neovascularization	Growth of new blood vessels	N/A (not a common feature)
Arteriolar Narrowing	N/A (not a common feature)	Retinal arteries appear narrower
AV Nicking	N/A (not a common feature)	Arteriovenous (AV) nicking
Flame-Shaped Hemorrhages	N/A (not a common feature)	Linear bleeding following retinal nerve fibers
Copper Wiring	N/A (not a common feature)	Thicker, less transparent retinal arteries
Papilledema	N/A (not a common feature)	Swelling of optic disc due to increased pressure

**Table 3 diagnostics-13-03236-t003:** A table summarizing the cited references, their methodologies, results, and limitations.

Cited	Methodology	Results	Limitations
[19]	Enhanced candidate areas using Gabor filter bank, followed by adaptive thresholding.	Detection of cotton wool patches (CWS) as significant diagnostic clues for HR.	No quantitative results or performance metrics mentioned.
[20]	Multi-layered neural network with invariant moment markers, Gabor coefficients, and wavelet coefficients.	Noticeably improved segmentation outcomes on the DRIVE dataset.	No specific quantitative results mentioned.
[21]	Automated nine-step process for OD region extraction, vessel segmentation, color feature detection, A/V ratio calculation, and HR classification.	AUC of 0.84, 90% sensitivity, and 67% specificity on 74 color fundus images.	Limited specificity and potentially complex nine-step process.
[22]	Independent component analysis (ICA) on wavelet sub-band to identify retinal changes.	Accurate identification of retinal changes on 50 photographs.	No specific quantitative results mentioned.
[23]	CLAHE applied for enhanced retinal vessel visibility, followed by morphological operations and neural network classification.	Classification accuracy of 95%.	Lack of detailed method description and potentially complex approach.
[24]	ELM classifier for retinal artery segmentation using local and morphological attributes.	A total of 96% accuracy, 71.4% sensitivity, and 98.6% specificity on the DRIVE dataset.	Specificity is high, but sensitivity could be improved.
[25]	Architecture combining random Boltzmann machine (RBM) and deep neural network (DNN) techniques for arterial blood vessel changes.	Significantly improved accuracy compared to previous methods.	No specific quantitative results mentioned.
[26]	CNN-based technique for extracting optic disc, retinal arteries, and fovea centralis.	Mean classification accuracy of 92.6% on the DRIVE dataset.	No specific sensitivity, specificity, or detailed results mentioned.
[27,28,29]	CNN-based methods for segmenting retinal vasculature into arteries and veins.	High accuracy (88.9% to 93+%) on different datasets.	No specific quantitative results for each reference mentioned.
[30]	Automatic CNN method for exudate detection in retinography images.	A 77% F-score and comparable performance metrics on the DRiDB dataset.	No specific sensitivity, specificity, or detailed results mentioned.

**Table 5 diagnostics-13-03236-t005:** Details of blocks and layers utilized for development of HDR-EfficientNet architecture.

Proposed Architecture Layers
Parameter for BN = BatchNormalization
Parameter for Convolution block = Convolution-filter (3 × 3)
Parameter for Convolution block 1: 2 × FMBConv1 Block
Parameter for Convolution block 2: 3 × FMBConv4 Block
Parameter for Convolution block 3: 2 × FMBConv4 Block
Parameter for Convolution block 4: 5 × MBConv4 Block
Parameter for Convolution block 5: 7 × MBConv6 Block
Parameter for Convolution block 6: 12 × MBConv6 Block
Additional Block: Conv2d (1 × 1) Block
Add function GAP: Global average pooling
Dense and Dropout layer
Prediction layer: FC Layer and Softmax layer

**Table 6 diagnostics-13-03236-t006:** Specifics of image processing-based data augmentation that was implemented.

Augmented Parameters	Optimal Values
Parameter 1: Affine transform	Assigned 1: True
Parameter 2: Pan	Assigned 2: True
Parameter 3: Spin-range	Assigned 3: 0.2
Parameter 4: Crop	Assigned 4: True
Parameter 5: Horizontal-flip	Assigned 5: True
Parameter 6: Vertical-flip	Assigned 6: False
Parameter 7: Affine transform	Assigned 7: True

**Table 7 diagnostics-13-03236-t007:** Average processing time on a HR and DR datasets by various DL algorithms.

Method	Preprocessing	Feature Extraction	Training	Prediction	Overall
VGG16	20.5 s	14.4 s	200.5 s	10.8 s	246.2 s
AlexNet	18.6 s	12.2 s	190.5 s	8.8 s	230.1 s
InceptionV3	16.3 s	14.8 s	178.5 s	7.8 s	217.4 s
GoogleNet	17.2 s	17.3 s	170.5 s	6.8 s	211.8 s
Xception	18.1 s	15.1 s	165.5 s	8.8 s	207.5 s
MobileNet	14.1 s	13.3 s	160.5 s	7.8 s	195.7 s
SqueezeNet	10.8 s	8.3 s	168.5 s	5.8 s	193.4 s
HDR-EfficientNet	1.8 s	1.9 s	165.5 s	1.5 s	184.5 s

**Table 9 diagnostics-13-03236-t009:** Assessment of the performance comparison based on classification accuracy of the deep learning arts with the proposed model.

State-of-the-Art Models	Augment	Time (S)	ACC
RBM-DNN [25]	Yes	38	79.1%
CNN-technique [26]	Yes	55	83.5%
EfficientNet [31]	Yes	35	78.84%
ResNet50 [23]	Yes	42	73.75%
DenseNet-101 [24]	Yes	39	93.93%
Proposed HDR-EfficientNet	Yes	30	98.12%

**Table 10 diagnostics-13-03236-t010:** Results of the proposed system model’s classification using 32 batches of data.

Model	Epochs	* SE	* SP	* ACC	* PR	F1-Score	* MCC
VGG16	40	78	80	79	76	79	80
AlexNet	40	79	82	81.1	80	80.0	81.0
InceptionV3	40	81	80	82.3	82	82.2	83.4
GoogleNet	40	83	81	83.6	83	83.3	84.3
Xception	40	82	83	82.6	83	84.4	85.2
MobileNet	40	84	84.0	84.3	84	85.1	86.1
SqueezeNet	40	85	86.1	87.2	85	86.0	87.0
HDR-EfficientNet	40	94	96	95.6	94.12	95.2	96.7

* SE: Sensitivity, SP: Specificity, PR: Precision, ACC: Accuracy, MCC: Matthews correlation coefficient.

**Table 11 diagnostics-13-03236-t011:** Results of the proposed system model’s classification using 64 batches of data.

Model	Epochs	* SE	* SP	* ACC	* PR	F1-Score	* MCC
VGG16	40	78	80	80	76	79	80.5
AlexNet	40	80	81	80.3	80	80.4	82.3
InceptionV3	40	82	82	81.7	82	82.7	84.0
GoogleNet	40	82	83	82.5	83	82.5	85.0
Xception	40	84	84	83.4	83	83.3	86.0
MobileNet	40	83	82.2	85.3	84	84.2	83.0
SqueezeNet	40	85	85.2	86.6	85	85.1	86.1
HDR-EfficientNet	40	94	96	95.6	94.12	95.2	96.7

* SE: Sensitivity, SP: Specificity, PR: Precision, ACC: Accuracy, MCC: Matthews correlation coefficient.

**Table 12 diagnostics-13-03236-t012:** Limitations of current system and future research are described.

Limitations	Future Research
Limited dataset size: The system’s performance may be constrained by the size and diversity of the available dataset.	Dataset expansion: Collect a larger and more diverse dataset to enhance the model’s ability to generalize across different populations and variations.
Class imbalance: Imbalanced class distributions can lead to biased learning and reduced accuracy for minority classes.	Data augmentation: Implement advanced data augmentation techniques to balance class distributions and improve classification accuracy for underrepresented classes.
Overfitting: Deep learning models, especially with complex architectures, can be prone to overfitting, leading to poor generalization on unseen data.	Regularization techniques: Explore various regularization techniques such as dropout, L1/L2 regularization, and early stopping to mitigate overfitting and enhance model robustness.
Computational resources: Deep learning models, particularly those with advanced architectures, can demand substantial computational resources for training and inference.	Model compression: Investigate model compression techniques to reduce model complexity while preserving performance, enabling deployment on resource-constrained devices.

## Data Availability

Dataset will be available upon request.

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
