# Peer review of "HDR-EfficientNet: A Classification of Hypertensive and Diabetic Retinopathy Using Optimize EfficientNet Architecture"

_diagnostics, 2023, doi:10.3390/diagnostics13203236_

Round 1
Reviewer 1 Report (Previous Reviewer 2)
Thanks for addressing my comments, but some issues need to be considered as well such as:
Please make sure that abbreviations are defined the first time they appear, then afterwards use these abbreviations not the full wording.
English need to be revised.
Figure 9 is confusing, why there are lines and bars?
Some English and spelling mistakes are detected
Author Response
Original Manuscript ID: ID: diagnostics-2597992
Original Article Title: HDR-EfficientNet: A Classification of Hypertensive and Diabetic Retinopathy Using Optimize EfficientNet Architecture
To: Editor in Chief,
MDPI, Diagnostics
Re: Response to reviewers
Dear Editor,
Many thanks for insightful comments and suggestions of the referees. Thank you for allowing a resubmission of our manuscript, with an opportunity to address the reviewers’ comments.
We are uploading (a) our point-by-point response to the comments (below) (response to reviewers), (b) an updated manuscript with green, blue, and orange highlighting indicating changes, and (c) a clean updated manuscript without highlights (PDF main document).
By following reviewers’ comments, we made substantial modifications in our paper to improve its clarity, English and readability. In our revised paper, we represent the improved manuscript such as:
(1) Revised Abstract, (2) Revised Introduction, (3) Results section, (4) Discussions and Conclusion sections.
We have made the following modifications as desired by the reviewers:
Best regards,
Corresponding Author,
Dr. Qaisar Abbas (On behalf of authors),
Professor.

Reviewer 2 Report (New Reviewer)
This paper is extremely long. Consider eliminating much of the background details including justifications etc. The paper should end up no more than 10 pages excluding references. Many of the comments are aimed at reducing the size of the document. The English requires extensive modification to make it clear to read. I have made some comments indication possible changes, however the document is so large that it is not possible to completely review and suggest changes for the rest of it. There is quite a bit of repetition as well. Some of the sections are muddled so for example the results section contains parts of setup as well as results. The computational gains of the authors’ methods are not highly significant. Most readers are more interested in the accuracy and other aspects of the new technique. Some of the details must be abbreviated. While interesting they are too extensive to put into a journal article.
Ultimately I think this is a useful paper but it needs some work to make it shorter and hit the relevant points better.
Line |
|
72-76 |
Needs to be rewritten. HR does not cause mortality itself and hypertension is not an eye disease but may cause an eye disease. Maybe you mean something like this:
Hypertension caused by an increase in artery pressure [4] and can harm several human organs, including the kidneys, heart, and retina [5]. |
77 |
What is HE? Cotton wool patches? Is that something visible on eye examination? |
79 |
How can treatment save lives? I understand it can save vision but explain briefly “proper treatment of ocular disorders associated with HR can save human lives [7] by … Also not necessary to specify “human”. |
86 |
Use “disease” rather than “sickness” |
87 |
“The extensive retinal veins lower the A/V ratio and are a vital HR indicator (the mean artery-to-vein diameter).” Is confusing. Do you mean something like “In HE, the increased presence of retinal veins lowers the artery-to-vein diameter (i.e. the A/V ratio, where A is the total diameter of all arteries and V is the total diameter of all veins in the image) and is an important HR indicator” |
90 |
Is AVR the same as A/V? If so be consistent. Pick one. |
93-94 |
Confusing and incorrect. Do you mean “These anomalies are indicative of damage to essential parts of the eye and may escalate to HR if they are not caught in time.” |
95 |
Replace “It was discovered that” with “In addition”. Also add “be” at the end of the sentence “…may be [11].” |
97 |
I’m not sure what you mean by “self-diagnosis”. Does the patient diagnose himself? Maybe you mean “immediate diagnosis”? |
100 |
Which authors offered the review? Need a reference. If it is you doing the review in this paper, then it is maybe best if you leave this line out entirely. |
110 |
Maybe replace with “EfficientNet incorporates attention mechanisms that enhance its ability to capture subtle features within medical images such as retinography scans, which is crucial for accurate disease detection.” |
Fig 1 |
The micro-aneurisms are not really visible. Do you have a better photo or maybe have show a close-up of that area as an inset in the figure? What are image b and c? Do they show some pathology or are they just general images? Need to put in the caption. |
Clinical Importance |
Table 1 should be summarized into just a few lines. |
Research Highlights |
This is not a standard section in a paper. Consider putting this content in the discussion or conclusion section. |
Research outline |
Also not standard. Consider eliminating completely. |
Background |
I like the first part of the background section and the part dealing with the differences between DR and HR, you should consider eliminating the Literature review except maybe leaving table 3. Normally such an extensive review is not required, just a few highlights. |
338 |
I am confused what images you used. Where did the 7400 images come from? Did you also use the data sets of Table 4? This may become clear later but here its confusing. |
Experimental Setup |
Should be in Materials and Methods not results. Same for all the other sections up to Results Analysis. |
Computational cost |
Should be mostly eliminated. Maybe just the table and a few words. It’s not that significant a difference between the methods. |
Limitations |
There is a lot in here but we only need a short |
References |
There are a lot of references for an original article – normally we would only see this many for a review article. It indicates that the sections on previous work are probably bloated. |
|
|
|
|
Addressed in the table in the comments section. Consider getting a native English speaker to review before re-submission,
Author Response
Original Manuscript ID: ID: diagnostics-2597992
Original Article Title: HDR-EfficientNet: A Classification of Hypertensive and Diabetic Retinopathy Using Optimize EfficientNet Architecture
To: Editor in Chief,
MDPI, Diagnostics
Re: Response to reviewers
Dear Editor,
Many thanks for insightful comments and suggestions of the referees. Thank you for allowing a resubmission of our manuscript, with an opportunity to address the reviewers’ comments.
We are uploading (a) our point-by-point response to the comments (below) (response to reviewers), (b) an updated manuscript with green, blue, and orange highlighting indicating changes, and (c) a clean updated manuscript without highlights (PDF main document).
By following reviewers’ comments, we made substantial modifications in our paper to improve its clarity, English and readability. In our revised paper, we represent the improved manuscript such as:
(1) Revised Abstract, (2) Revised Introduction, (3) Results section, (4) Discussions and Conclusion sections.
We have made the following modifications as desired by the reviewers:
Best regards,
Corresponding Author,
Dr. Qaisar Abbas (On behalf of authors),
Professor.

Round 2
Reviewer 1 Report (Previous Reviewer 2)
Thanks for addressing my comments
This manuscript is a resubmission of an earlier submission. The following is a list of the peer review reports and author responses from that submission.
Round 1
Reviewer 1 Report
The manuscript presented a deep learning approach (HDR-EfficientNet) to classify hypertensive retinopathy (HR) and diabetic retinopathy (DR) based on retinal fundus images. These two diseases HR and DR present some similar features on the fundus images, it would be helpful to differentiate them using an automated approach. However, the manuscript was poorly presented with numerous places of errors and inconsistencies. In addition, this reviewer has major concerns as follows:
1. Based on the example image, HR is in the severe stage with disc edema whereas the DR is in the mild stage. Such classification seems to lack significance since their differences are so obvious without a need of a complicated model. The authors discussed the importance of early detection of HR while using late-stage HR images as training data. The purpose of the study did not align with the approach.
2. What is the source of the image dataset?
3. It is unclear whether the authors used the existing EfficientNet model. If so, what is the novelty of the methodology?
4. Figure 7 needs clarification
Extensive editing of English language is required
Reviewer 2 Report
The paper introduces a deep learning (DL) method called HDR-EfficientNet, which aims to provide an efficient and accurate approach for identifying various eye-related disorders, including diabetes and hypertensive retinopathy. The proposed method utilizes an EfficientNet-V2 network for end-to-end training focused on disease classification. Additionally, a spatial-channel attention method is incorporated into the approach to enhance its ability to identify specific areas of damage and differentiate between different illnesses. The HDR-EfficientNet model is developed using transfer learning, which helps overcome the challenge of imbalanced sample classes and improves the generalization of the network. Dense layers are added to the model structure to enhance the feature selection capacity. The topic is interesting. However, some comments need to be addressed before acceptance. Also, the literature review section misses some important papers.
Introduction
There are many deep convolutional neural network (DCNN) architectures that are lightweight please explain why did you use EfficientNet.
Could you please explain the novelty in the introduction?
Section 2 contains some basic information and background, could you please remove it? You can only keep the information regarding Efficient Net and attention.
Literature Review
Some important papers are missing related to diabetic retinopathy diagnosis using deep learning methods. Could you please add the following articles?
These articles use DR diagnosis
GabROP: Gabor Wavelets-Based CAD for Retinopathy of Prematurity Diagnosis via Convolutional Neural Networks
DIAROP: Automated Deep Learning-Based Diagnostic Tool for Retinopathy of Prematurity
Detection and classification of red lesions from retinal images for diabetic retinopathy detection using deep learning models
A Lightweight Robust Deep Learning Model Gained High Accuracy in Classifying a Wide Range of Diabetic Retinopathy Images
An active deep learning method for diabetic retinopathy detection in segmented fundus images using artificial bee colony algorithm
A novel approach for diabetic retinopathy screening using asymmetric deep learning features
Methodology:
Lines 294-298 should be moved to the conclusion section.
How did you ensure that augmented versions of the image is not located n both testing and training sets?
Please add the number of images in each class.
HDR-EFFICIENTNET is sometimes written in small letters and other places in capital. Please check
Please explain the difference between that Efficient Net you used and previous versions.
What is the learning rate used? How did you choose the values of the hyperparameters?
Experimental Results
Please validate the performance of the proposed framework on one more dataset.
Specificity matric definition and equation are not included, however, it is used in the results. Please add. Also please mention that recall is also called sensitivity.
Figure 7 resolution is very poor.
There are two figure 7. Please check and correct.
Conclusion
The section is too long. Please summarize it.
Reviewer 3 Report
This paper can be accepted directly.
No